# Spatial and Temporal Aspects of Fungicide Resistance in *Venturia inaequalis* (Apple Scab) Populations in Northern Germany

**DOI:** 10.3390/biotech14020044

**Published:** 2025-06-05

**Authors:** Roland W. S. Weber, Rebekka Busch, Johanna Wesche

**Affiliations:** 1Lower Saxony Chamber of Agriculture, Esteburg Centre, Moorende 53, 21635 Jork, Germany; 2Department of Food Science, Aarhus University, Agro Food Park 48, 8200 Aarhus, Denmark; 3Independent Researcher, Formerly Department of Biology, University of Hamburg, Ohnhorststr. 18, 22609 Hamburg, Germany; rebekka.busch@web.de; 4Department of Plant and Environmental Science, Clemson University, Clemson, SC 29630, USA; jwesche@clemson.edu

**Keywords:** Altes Land, anilinopyrimidines, apple scab, dodine, IPM, Lower Elbe region, MBC, population dynamics, QoI, SDHI

## Abstract

*Venturia inaequalis*, the cause of apple scab, readily develops resistance to fungicides with specific modes of action. Knowledge of the spatial and temporal pattern of resistance development is therefore relevant to fruit producers and their consultants. In the Lower Elbe region of Northern Germany, a two-year survey based on a conidial germination test was conducted, examining fungicide resistance in 35 orchards under Integrated Pest Management (IPM), 16 orchards of susceptible cultivars as well as a further 12 orchards of scab-resistant (*Vf*) cultivars under organic management, and 34 abandoned or unmanaged sites. No evidence of resistance to SDHI compounds (fluopyram, fluxapyroxad) was found after >5 yr of their regular use. Resistance to anilinopyrimidines (cyprodinil, pyrimethanil) had disappeared 15 yr after its widespread occurrence. Isolates from a few IPM orchards showed a reduced sensitivity to dodine. Double resistance to the MBC compound thiophanate-methyl and the QoI trifloxystrobin was rare in *V. inaequalis* strains that had achieved breakage of *Vf*-resistance, but very common (>50%) on scab-susceptible cultivars in IPM, organic and abandoned orchards in the ‘Altes Land’ core area of the Lower Elbe region, and in IPM orchards in the periphery. We conclude that resistance to QoI and MBC fungicides is persistent even decades after their last use, and that the core area harbours a uniform population adapted to intensive crop protection, whereas isolated orchards in the periphery are colonised by discrete populations of *V. inaequalis*.

## 1. Introduction

*Venturia inaequalis* (Cke.) Wint., the cause of apple scab, is the most important fungal pathogen of apple production worldwide [1,2]. This disease is most prominent in regions with a humid and mild spring climate, including northwestern Europe. Scab lesions are common on foliage as well as fruit. In the Lower Elbe region, which is Germany’s largest apple-growing area with a total annual yield exceeding 300,000 t [3], fruit losses of up to 10% or 50% are possible in orchards under integrated pest management (IPM) or organic management, respectively, if seasonal conditions favour infections (Figure 1). In both production forms, well over one-half of all crop protection measures of the year are directed solely against apple scab (R. Weber, unpublished data). Scab-resistant (*Vf*) cultivars would seem to offer potential savings of fungicide applications. However, such savings are compromised by the rise in other diseases, such as sooty blotch, in the absence of regular scab sprays [4]. Furthermore, *V. inaequalis* is capable of overcoming such monogenetic resistance especially in situations of reduced fungicide input [5].

Following the maturation of sexual reproductive structures (pseudothecia) of *V. inaequalis* on fallen infected leaves during winter, ascospores are actively discharged from bud break until after flowering, initiating infections of unfolding leaves and fruitlets. Ascospores are airborne and may carry infections over distances of 25 m in practice [1,6], although individual spores may travel distances of several kilometres [1]. Within 2–4 weeks of infection, fresh scab lesions emerge, and these produce conidia on their surface which are dispersed by rain-splash, thereby spreading the infection mainly within an infected tree [1,7]. During wet summers, several cycles of conidial infections are possible. Therefore, fruit growers must focus on controlling ascospore infections in spring in order to be able to reduce fungicide input during summer.

In IPM, as well as organic farming, protectant (multi-site) fungicides should be applied before the onset of rainfall. Sprays within a few hours of incipient rain are also possible so long as the germ-tubes emerging from ascospores or conidia are still confined to the leaf surface. Following penetration of the leaf cuticle by the first germ-tubes, only compounds with a curative activity may be sprayed to control an ongoing infection. These are mostly single-site compounds with specific modes of action, making them susceptible to fungicide resistance development by the fungus.

Along with *Botrytis* spp. [8,9], *V. inaequalis* is a known high-risk pathogen of horticultural crops with respect to fungicide resistance [10,11]. Most curative fungicide classes have been rendered ineffective by the rapid spread of resistant strains, often within a few years of usage. For example, in the Lower Elbe region the successive occurrence of resistance events, each time associated with a commercially serious loss of efficacy of the relevant fungicides, has been recorded for methyl benzimidazole carbamates (MBCs) in 1974 [12], triazoles in 1996 [13], QoI fungicides in 1998 [14] and anilinopyrimidines (APs) in 2005 [15], i.e., within 3, 15, 2 and 9 years of the beginning of their widespread commercial usage, respectively. This pattern of a successive emergence of resistance as recorded in the Lower Elbe region appears to be typical of worldwide trends [16,17,18].

In some countries such as the United States, dodine, a compound commonly deployed as a protectant fungicide from the 1960s onwards, has been subject to resistance development after being heavily used for many successive years [17,19]. In Germany, dodine was registered for 10 years from 1961 and thereafter remained unavailable until the 2008 season [20]. In the Lower Elbe region, it was never recommended for major use as a protectant fungicide, but it has been deployed more sparingly in a curative context after the discovery of its relevant properties [20]. A regional survey in 2014 failed to detect any evidence of resistance to dodine [21].

Single-site fungicides of the succinate dehydrogenase inhibitor (SDHI) group were introduced into Northern German fruit production after the turn of the millennium, and for control of scab and powdery mildew (*Podosphaera leucotricha* (Ellis & Everh.) E.S. Salmon) in pome fruit in 2013. From the beginning, they have been deployed in a purely protectant function, being applied in a tank mixture with a multi-site compound such as captan or dithianon ahead of severe expected infections [22]. Few reports of SHDI resistance exist in *V. inaequalis* worldwide [11,23].

Monitoring of *V. inaequalis* resistance to the commonly used single-site fungicides should be conducted by any responsible advisory service in order to detect incipient resistance development and adjust spray recommendations in good time to prevent the spread of resistant strains. Here, we report the results of an extensive survey conducted during the 2021 and 2022 seasons in the Lower Elbe region, focusing on scab-relevant AP, QoI and SDHI fungicides as well as dodine. In addition, since we were interested in the persistence of fungicide resistance over time, we included in our survey a representative of MBCs which ceased to be used against scab in 1975 [12], while their sporadic pre-harvest application against fungal storage rots continued until 2006 [24]. Finally, we wanted to know how far fungicide-resistant strains of *V. inaequalis* had spread from their putative centre of origin within the core area into the periphery of the Lower Elbe region.

## 2. Materials and Methods

### 2.1. Sampling Area

The Lower Elbe region, situated southwest of the city of Hamburg along the estuary of the river Elbe, comprises some 9500 ha of fruit trees, of which apple makes up a 91% share [25]. The core area is the Altes Land (53.49–53.62° N; 9.51–9.90° E) with a continuous fruit tree acreage of approx. 8500 ha [26], making it the second-largest coherent and the northernmost major fruit-growing area in Europe. For the purposes of this survey, the Altes Land was demarcated between the southern shore of the river Elbe to the north, the dam of the A26 motorway to the south, and the cities of Hamburg and Stade to the east and west, respectively (indicated in Section 3.3’). In the Altes Land, a long record of fruit cultivation stretching back as far as A.D. 1312 has been tied closely to a highly fertile soil and frost protection afforded by the presence of numerous open drainage ditches, both having their origin in the history of the area as marshland reclaimed from the shores of the Elbe river in the early middle ages [26]. Today, an effective frost protection is secured by overhead sprinkler irrigation systems which have been fitted to almost all commercial orchards.

More than 75% of the apple orchard area in the Lower Elbe region is managed according to IPM principles. Organic orchards currently comprise about 20% of the acreage, and they are interspersed among IPM orchards. Many organic farms have arisen by conversion from IPM predecessors, and these continue to grow highly scab-susceptible cultivars such as ‘Braeburn’, ‘Jonagold’ (including the variant ‘Red Jonaprince’), and ‘Elstar’. Scab-resistant cultivars comprise a minor but steadily growing share of the total organic acreage in the area. In 2019, this share was approaching 30% [27]. They are of no commercial relevance to IPM as yet.

Scattered across the Altes Land and the periphery of the Lower Elbe region, there are orchards planted for non-commercial purposes and others which have been abandoned but may continue to exist for decades. These were considered valuable for fungicide resistance studies in having been left unsprayed since their time of planting or abandonment, respectively. For the past 60 years, trees in commercial orchards have been planted as grafts on dwarf rootstocks (chiefly M9) at a distance of approx. 1 m within and 3.5 m between rows (Figure 2). However, older abandoned or non-commercial orchards often comprise trees on seedling rootstocks with larger canopies and correspondingly wider planting distances (Figure 3).

### 2.2. Reagents

Fungicides containing the following active ingredients were obtained as formulated commercial products: dodine as Syllit^®^ (UPL GmbH, Brühl, Germany); the QoI trifloxystrobin as Flint^®^ (Bayer CropScience, Monheim, Germany); the SDHIs fluopyram as Luna Privilege^®^ (Bayer CropScience, Monheim, Germany) and fluxapyroxad as Sercadis^®^ (BASF SE, Ludwigshafen, Germany) and the APs cyprodinil as Chorus^®^ (Syngenta Agro GmbH, Maintal, Germany) and pyrimethanil as Scala^®^ (BASF SE, Ludwigshafen, Germany). In addition, the MBC compound thiophanate-methyl was obtained as Cercobin FL^®^ (BASF SE, Ludwigshafen, Germany). These fungicides were prepared as aqueous stock suspensions or emulsions which were added to the cooling agar after autoclaving to give the final concentration of active ingredients as required. All other reagents were provided by Carl Roth (Karlsruhe, Germany).

### 2.3. Characterisation of Isolates for Fungicide Resistance

For detailed analyses of resistance properties, *V. inaequalis* was isolated by streaking aqueous suspensions of conidia from fresh scab lesions onto potato dextrose agar (PDA) augmented with 200 mg penicillin G and 200 mg streptomycin sulphate L^−1^. The antibiotics were added to the cooling agar after autoclaving. Pure cultures of *V. inaequalis* were obtained by repeated sub-cultivation onto PDA without antibiotics. Sporulation on PDA was promoted by exposing growing PDA cultures at room temperature to near-UV light (peak emission at 365 nm) for 15–30 min per day. Fresh conidia were harvested in 5–10 mL of a spore preservation medium (5 mL glycerol L^−1^ double-strength skimmed milk) and stored as lyophilised preparations in vacuum-sealed glass ampoules [28]. Details of all isolates used in this study are provided in Table 1.

The conidial germination assay previously developed for *Botrytis* spp. [29,30] was suitable also for *V. inaequalis*; a test for dodine was added to the portfolio, as described below. In order to encourage reliable conidium production in pure culture, 20 pieces (2–5 mm^2^) of colonies growing on 1% (*w*/*v*) malt extract agar (MEA) were excised, transferred to liquid 1% (*w*/*v*) malt extract medium, and incubated at room temperature in daylight for 7–21 d except for a daily exposure to near-UV light for 15–30 min. For resistance testing of pure-culture isolates, spores were directly harvested from the malt extract medium by pipetting.

MEA was used for thiophanate-methyl and dodine assays. For trifloxystrobin, the alternative oxidase inhibitor salicyl hydroxamic acid was added at a standard concentration of 100 mg L^−1^ MEA [29,31]. For cyprodinil and pyrimethanil assays, 0.5% (*w*/*v*) sucrose agar augmented with 0.05% (*w*/*v*) yeast extract was used instead of MEA, whereas for fluopyram and fluxapyroxad 0.5% (*w*/*v*) yeast extract agar was used [29,30]. In order to establish inhibition curves, a wide range of concentrations of active ingredients was tested (see Section 3.1). For each concentration and each fungicide, 17 µL aliquots of freshly harvested spore suspension were placed on the test agar media. Following an incubation of 24–30 h at 20 °C, 10 representative germ-tube lengths were measured for each fungicide concentration with a light microscope (Axio Lab A1; Carl Zeiss, Göttingen, Germany) at 100-fold final magnification. Each concentration series was measured in 2–3 independent assays. Effective concentrations at which germ-tube growth was inhibited by 50% (EC_50_) were calculated by regressing the germ-tube length as percent of growth relative to the fungicide-free control medium against log_10_ of concentrations of active fungicide ingredient.

### 2.4. Fungicide Resistance Screening

For most fungicides except dodine, two discriminatory concentrations in addition to the zero-fungicide control were chosen from the detailed inhibition curves to permit a distinction of sensitive from resistant responses in routine assays. Growth was categorised as full (>50% of control), inhibited (<50%) with or without hyphal deformations, stalled (no further growth after germ-tube emergence) or absent [29]. These discriminatory concentrations were 1 and 100 mg L^−1^ for thiophanate-methyl; 0.1 and 10 mg L^−1^ for trifloxystrobin; 1 and 25 mg L^−1^ for cyprodinil and pyrimethanil; and 1 and 50 mg L^−1^ for fluopyram and fluxapyroxad. For dodine, three concentrations (0.1, 0.2 and 0.5 mg L^−1^) were chosen and the length of 10 germ-tubes was measured as above in order to compare results with the data of Köller et al. [32]. Similarly, in light of previous results showing EC_50_ values of 0.2 mg L^−1^ for susceptible isolates but exceeding 3 mg L^−1^ for resistant ones [21], germ-tube length was measured for cyprodinil at 1 mg L^−1^.

Depending on availability of material, spore suspensions from at least 10 scab lesions per orchard were examined wherever possible. Conidia were removed from a young individual scab lesion in a drop of water using a micropipette, diluted in 1 mL water and plated out as 17 µL droplets onto each of the test plates. Up to 30 drops were accommodated per agar plate, permitting the screening of 2–3 orchards per set of test plates.

In the 2021 and 2022 seasons, we sampled a total of 35 IPM orchards, 16 organic orchards planted with standard (scab-susceptible) cultivars such as ‘Braeburn’, ‘Jonagold’ and ‘Elstar’, and a further 12 organic orchards containing scab-resistant (*Vf*) cultivars such as ‘Topaz’, ‘Antares/Dalinbel’, ‘Santana’ and ‘ZIN17/Deichperle’. In addition, from within the Altes Land and its periphery, samples were collected from 13 and 21 (respectively) abandoned orchards (no fungicide treatments for at least 10 years) or individual untreated apple trees. Such orchards or trees were identified by several criteria, including a lack of weed control; growth of perennial plants within and between the tree rows; lack of tree pruning leading to a crowded and aged canopy structure; excessive callus formation around canker infections by *Neonectria ditissima* (Tul. & C. Tul.) Samuels & Rossman indicating long-term lack of canker pruning; colonisation of the tree bark by a diversity of epiphytic lichens; and the presence of leaf-mining moths and/or shoot dieback due to *Monilinia fructigena* (Aderh. & Ruhl.) Honey, indicating the long-term absence of basic crop protection measures (Figure 4).

### 2.5. Data Analysis

All statistical analyses were performed using the SPSS 29.0 software (IBM, Amonk, NY, USA). To analyse for statistical differences between different production systems (IPM, organic and organic *Vf*) within in the Altes Land region and the periphery for the percentage of TM and TFS resistance, the non-parametric Kruskal–Wallis test (*p* < 0.05) was used. Significance values were adjusted using the Bonferroni correction for multiple comparisons.

For analyses of differences between the two regions (Altes Land and periphery) for the percentage of TM and TFS resistance in abandoned orchards, a Kruskal–Wallis test (*p* < 0.05) based on the medians was used.

No form of generative artificial intelligence was used at any stage during data generation, analysis and writing of this manuscript.

## 3. Results

### 3.1. Characterisation of Fungicide Resistance

Twenty *V. inaequalis* isolates obtained from 13 different orchards were characterised in detail for their response to the seven fungicides tested. The results are summarised as EC_50_ values in Table 1.

Highly diverse resistance responses were observed for thiophanate-methyl. EC_50_ values of sensitive isolates were below 0.11 mg L^−1^ whilst those of resistant ones were in the range of 1.7–25.1 mg L^−1^. At 1 mg L^−1^, sensitive isolates were readily recognised by their rudimentary growth as contorted, coiled germ-tubes whereas resistant ones displayed > 50% growth (Figure 5). At 100 mg L^−1^, only very few highly resistant strains such as isolate 45-29 displayed any residual growth (Figure 6).

Even more pronounced differences were recorded for the QoI fungicide trifloxystrobin (Figure 7). Whereas sensitive isolates gave rise to EC_50_ values of 0.002–0.01 mg L^−1^, highly resistant ones failed to be inhibited even at the highest concentration of 10 mg L^−1^. EC_50_ values for such isolates could not be calculated. At a discriminatory trifloxystrobin concentration of 0.1 mg L^−1^, susceptible isolates displayed no growth or only stalled germination, whereas resistant isolates showed uninhibited growth even at 10 mg L^−1^ (Figure 7).

In contrast to the above, dodine EC_50_ values for the 20 *V. inaequalis* isolates analysed in detail were in a relatively narrow range of 0.05–0.35 mg L^−1^ (Table 1). Nonetheless, inhibition curves of the least and most sensitive isolates displayed differences in the range of 0.1–0.5 mg dodine L^−1^ (Figure 8) which could be quantified by measuring germ-tube length at 0.2 mg L^−1^ in routine assays.

Relatively wide and continuous ranges of EC_50_ values (Table 1) were obtained for the AP compounds pyrimethanil (0.11–0.52 mg L^−1^) and cyprodinil (0.02–0.39 mg L^−1^). This was reflected by representative inhibition curves for cyprodinil which permitted only a vague differentiation between stronger or lesser growth at 1 mg L^−1^, although this concentration would have revealed resistant isolates with EC_50_ values exceeding 1 mg L^−1^ in our survey (Figure 9). Individual isolates showed a fair correlation between their EC_50_ values for both fungicides (R^2^ = 0.393; Figure 10). Therefore, one of them—cyprodinil—was chosen for the resistance survey.

When tested against the two SDHI compounds fluopyram and fluxapyroxad, none of the 20 isolates examined in detail showed any evidence of resistance development, narrow EC_50_ ranges of 0.02–0.09 mg L^−1^ (fluopyram) and 0.002–0.07 mg L^−1^ (fluxapyroxad) being recorded. The inhibition curves showed no clear distinction between greater or lesser growth inhibition at 0.1 mg L^−1^ (Figure 5 and Figure 11). Due to a moderate correlation between the EC_50_ values of both compounds (R^2^ = 0.601; Figure 12), one of them—fluopyram—was used as a standard for the resistance survey. We assumed that any isolate with strongly reduced SDHI sensitivity would have been detected in this survey by germ-tube growth at 1 mg fluopyram L^−1^.

### 3.2. Fungicide Resistance in Commercial Orchards Within the Lower Elbe Region

In the 2021 and 2022 seasons, altogether, 380 isolates from 35 orchards under IPM, 169 isolates from 16 orchards planted with standard cultivars under organic management and a further 117 isolates from 12 orchards of scab-resistant (*Vf*) cultivars under organic management were tested for their sensitivity to thiophanate-methyl, trifloxystrobin, dodine, cyprodinil and fluopyram. Due to the different kinds of resistance response as described above, different forms of presenting the data for these fungicides are deployed here.

In the case of thiophanate-methyl and trifloxystrobin, resistance responses of a qualitative nature were recorded in the routine tests. In IPM orchards, resistance to thiophanate-methyl was universal, being recorded in every orchard sampled, and comprising 91.5% and 99.6% of all isolates in 2021 and 2022, respectively. At 92.9% and 92.4%, respectively, the corresponding values for trifloxystrobin were of a similar order. Most strains from IPM orchards therefore possessed double resistance to both compounds. This applied to the Altes Land core area as well as to the periphery (Figure 13). In organic orchards of the same scab-sensitive cultivars as grown in IPM—chiefly ‘Braeburn’, ‘Jona-gold’ and ‘Elstar’—a broadly similar picture was obtained, except that resistance to thio-phanate-methyl (88.5% in 2021, 76.1% in 2022) was more common than resistance to trifloxystrobin (28.2% and 31.5%, respectively). Moreover, organic orchards in the periphery harboured lower shares of resistance to either or both fungicides than those in the Altes Land (Figure 13). In marked contrast, the great majority (88.2% and 89.2% in 2021 and 2022, respectively) of *V. inaequalis* strains that had overcome the monogenetic *Vf* resistance of cultivars such as ‘Santana’, ‘Topaz’, ‘Antares/Dalinbel’ and ‘ZIN17/Deichperle’ were sensitive to both fungicides, with strains being resistant to either or both being observed in only 4 of the 12 orchards surveyed. The share of fungicide-resistant isolates on *Vf* cultivars was even lower in the periphery than in the Altes Land core area (Figure 13).

In the case of dodine, less pronounced differences made it necessary to quantify the resistance response for each isolate by determining its length of germ-tubes at 0.2 mg L^−1^ relative to the fungicide-free control. The results are presented in Figure 14 as combined data for the entire Lower Elbe region (Altes Land and periphery). A total of 11.8% of strains from IPM orchards (13.2% from Altes Land, 7.1% from periphery) showed at least 40% germ-tube growth at 0.2 mg dodine L^−1^, and these shares were unevenly distributed between individual orchards, ranging from 0% to 75%. More strains with elevated growth at 0.2 mg dodine L^−1^ were retrieved from IPM orchards known or strongly suspected to have received repeated dodine applications than in other IPM orchards where dodine had been used only 1–2 times per season (Figure 15). In contrast, none of the isolates from organically managed standard or *Vf* cultivars was able to show such elevated growth at 0.2 mg dodine L^−1^. The overall germ-tube length at 0.2 mg dodine L^−1^ was 20.4% ± 15.4% (average ± standard deviation) for strains from IPM orchards, whereas in the case of organic orchards these values were 9.9% ± 6.6% for standard cultivars and 12.2% ± 5.6% for *Vf* cultivars. None of the isolates analysed in the survey showed more than 50% growth at 0.5 mg dodine L^−1^.

As for dodine, a quantification of resistance response was attempted for cyprodinil by comparing the germ-tube length at 1 mg L^−1^ with the fungicide-free control. Data are presented together for the Altes Land and the periphery (Figure 16). Only a few isolates showed elevated (>60%) growth in IPM (1.9%) or in organic farms on standard cultivars (1.8%) or *Vf*-resistant cultivars (0.9%). Similarly, at 24.8% ± 13.9%, 26.7% ± 10.9% and 26.9% ±13.4% (respectively), the average germ-tube lengths at 1 mg cyprodinil L^−1^ were similar in all three categories.

No *V. inaequalis* isolate in the entire survey showed any obvious conidial germination or germ-tube growth at fluopyram concentrations of 1 or 50 mg L^−1^.

### 3.3. Regional Distribution of Fungicide Resistance in Abandoned Orchards

We also examined resistance to thiophanate-methyl and trifloxystrobin in *V. inaequalis* isolates from leaves collected from abandoned orchards or trees that had been left free from any chemical crop protection measure for at least 10 years. Sites from within the Altes Land core area as well as the open landscape of its periphery were sampled (Figure 17). The levels of resistance to both fungicides were significantly higher within the Altes Land than in its periphery (*p* < 0.05). In fact, all sites within the Altes Land yielded > 90% of isolates with resistance to at least one of the two fungicides, whereas only 5 of 21 locations in the periphery harboured > 50% resistance. The latter are labelled in Figure 17. Both location #1 (70-yr old ‘Boskoop’ trees on seedling rootstock) and location #2 (30-yr old trees of different cultivars on M9 rootstock) had been left untreated for 50 or at least 15 years, respectively, whereas locations #3 and #4 comprised small groups of younger (approx. 10 yr old) trees in a non-commercial context. Location #5 harboured a solitary roadside tree (>50 yr old) near Winsen in the far east of the sampling area. This tree had been previously sampled for QoI resistance in 2003 [33].

These striking differences between the Altes Land and its periphery with respect to fungicide resistance in abandoned orchards also became apparent in the pooled data sets (Figure 18) and were statistically significant (*p* < 0.05) both for MBC and QoI resistance.

## 4. Discussion

The survey presented here for *V. inaequalis* in the Lower Elbe region has revealed strains with resistance to MBC and QoI fungicides to be almost universally present in IPM orchards and also very frequent on standard apple cultivars in organic orchards, but rare in *Vf*-resistant apple cultivars under organic management. Furthermore, resistance to these fungicide groups was very common in abandoned or otherwise unmanaged orchards within the Altes Land, which has been the core area of fruit production in the Lower Elbe region for centuries, but rare or absent from similarly neglected but isolated orchards in the periphery of the region. In contrast to QoI and MBC resistance, signs of a weak shifting towards resistance to dodine were confined to IPM orchards, whereas no evidence of resistance to AP- and SDHI-type fungicides was recorded in any kind of orchard.

Our baseline EC_50_ values for the MBC compound thiophanate-methyl below 0.11 mg L^−1^ and for the QoI fungicide trifloxystrobin below 0.01 mg L^−1^ correlated well with other studies [18,34]. Likewise, full growth in routine assays at 1–5 or 2 mg L^−1^ (respectively) was indicative of resistance to these fungicides in our study as well as in the literature [11,16,35]. The point mutations responsible for MBC and QoI resistance have been well characterised at the molecular level, being located in the β-tubulin gene and the cytochrome b gene, respectively [34,35]. Both types of resistance are known to cause a high or even complete loss of efficacy of the fungicide groups concerned, but neither has been associated with any fitness deficit in past studies [36,37,38].

In contrast to MBCs and QoIs, the other three fungicide groups examined in our study are still in current use for scab control. Resistance to SDHIs such as boscalid and fluopyram is very widespread in Northern German *Botrytis* populations causing grey mould in soft and stone fruit production [30,39], and it has been reported from *Botrytis* spp. worldwide [40,41]. This is in contrast to *V. inaequalis* where evidence of resistance has so far been found mainly for boscalid but not for other SDHIs [11,42]. In our survey, we detected no evidence of any reduced sensitivity to fluopyram and fluxapyroxad in the Lower Elbe region, our EC_50_ values being within the published range of fully sensitive isolates [23,42,43]. Whilst boscalid has never been used for scab control in the Lower Elbe region, fluopyram and fluxapyroxad have been recommended since the 2015 season in tank mixtures with dithanon- or captan-based fungicides ahead of severe expected scab infections, and are being used by IPM farmers, typically 1–3 times per season [22] (R. Weber, unpublished). When applied in this way, SDHI fungicides also possess a high efficacy against powdery mildew. All available evidence suggests that the deployment of SDHIs in this manner is a sustainable strategy for resistance management.

In the Lower Elbe region, AP fungicides played an important role in scab control from the 1997 season until the detection of widespread resistance in 2005, when their use was terminated abruptly and completely [15,44]. Resistant isolates with EC_50_ values > 3 mg L^−1^ were still detected in our previous survey in 2014 [21], but not in 2021 and 2022 (current work). The EC_50_ values for pyrimethanil as determined in the present study mostly fell within, or only slightly exceeded, the baseline range of 0.12–0.3 mg L^−1^ reported by Köller et al. [45]. The registration of Faban^®^, a formulated mixture of dithanon and pyrimethanil, led us to conduct field trials which repeatedly showed a high efficacy against apple scab. In consequence, we have been recommending the use of Faban^®^ from the 2022 season onwards [46]. In agreement with the manufacturer, this fungicide is used up to three times per season in a strictly protectant manner (R. Weber, unpublished). Future fungicide resistance surveys will show whether this is sufficiently restrictive to prevent the re-emergence of AP resistance.

Following the failure of triazoles, APs and QoIs, no fungicide with a curative mode of action was temporarily available in the Lower Elbe region from 2005 until regional researchers characterised curative effects of dodine [20], leading to the recommendation of dodine from the 2010 season onwards in this way. The typical use of dodine has been approximately two times *per annum* according to recommendations (ranging from 1 to 3 times, depending on the season) as a curative measure during prolonged wetness periods about 2–4 days after a protectant spray [22]. The 2014 survey failed to provide any evidence of resistance development [21] whilst the data for 2021 and 2022 have revealed low proportions of strains capable of elevated growth at 0.2 mg L^−1^ dodine (present work). These were predominantly found in a few IPM orchards in which dodine had been used more frequently than recommended, and in a different way, viz. as a substitute, rather than a supplement, to a preceding protectant spray. In contrast, strains with enhanced growth at 0.2 mg dodine L^−1^ were entirely absent from organic orchards. These observations support recommendations concerning restrictions to the use of dodine in IPM. On the other hand, fully resistant populations such as those reported from the United States consist predominantly of strains capable of >80% growth at 0.2 mg dodine L^−1^ [32]. Only one strain of this calibre was detected in the Lower Elbe region. Therefore, whilst we cannot rule out the beginnings of a slight shift towards reduced sensitivity in the Lower Elbe region, the situation is not comparable to regions with abundant resistance following the repeated use of dodine over many years [32]. This is also supported by the range of EC_50_ values in our study which, at 0.05–0.35 mg L^−1^, corresponded to the lowest EC_50_ values recorded for New Zealand [47,48] and was also within the range of 0.02–1.2 mg L^−1^ described for baseline isolates from the United States [32]. Further, good scab control with dodine has been reported for orchards harbouring *V. inaequalis* strains with EC_50_ values below 0.65 mg L^−1^ [49] and poor control at EC_50_ values above 0.7 mg L^−1^ [50]. All of our 666 isolates tested in 2021 and 2022 showed < 50% germ-tube growth at a dodine concentration of 0.5 mg L^−1^, indicating EC_50_ values below this concentration and thereby supporting the notion that the Northern German population as a whole is as yet sensitive to this crucial fungicide.

Turning from practical implications of fungicide resistance surveys to more fundamental aspects, we note that strains with resistance to MBC and/or QoI fungicides may comprise the vast majority of a *V. inaequalis* population, as previously described in the literature [17,21,51]. In our surveys, the double-mutant phenotype was by far the most common, even in abandoned orchards within the Altes Land core area, indicating a high competitive fitness even in the long-term absence of fungicide treatments. In fact, we specifically re-visited the same unmanaged roadside apple tree at an isolated location in the periphery (Figure 13, location #5) where the QoI-type resistance had been found in 2003 [33]. The tree had been unmanaged back then and has remained so ever since, yet QoI resistance was found in 70% of *V. inaequalis* isolates from that site in 2022. All QoI-resistant strains were also resistant to thiophanate-methyl (not examined in 2003, but very probably already present). This is a particularly striking demonstration of the high persistence of fungicide-resistant *V. inaequalis* strains at the level of individual locations. To our knowledge, there are no previous reports of a similar durability of QoI and MBC resistance in *V. inaequalis* in habitats never subjected to any kind of selective pressure by fungicide use.

Whilst the persistence of strains harbouring QoI and MBC resistance mutations is obviously high, their mobility does not appear to be so. Thus, another apple tree group about 1 km north of location #5 failed to produce any QoI or MBC resistance. More strikingly, unmanaged apple trees in the periphery of the Altes Land—notably from 1 km to about 25 km to the south—were mostly free from MBC- and QoI-resistant *V. inaequalis* strains even 50 or 25 years (respectively) after their documented origin in the Altes Land [12,14]. Among the five exceptions to this generalisation, two locations comprised relatively young trees (approx. 10 years old) which might have carried resistant strains with them from the nurseries. This low overall proportion of strains with double resistance to MBCs and QoIs in unmanaged apple trees was in contrast to their high frequency in IPM orchards within the periphery.

These observations lead us to conclude that strains possessing fungicide resistance to MBCs and QoIs have effectively colonised the entire fruit production area of the Altes Land, but have not achieved a similar penetration in the periphery where individual orchards are often separated by distances of 1 km or more. A similar observation has been made with respect to the high frequency of strains with resistance to MBC fungicides within an intensive apple-growing area in the United States, and their absence from the periphery [52]. A landscape with isolated orchards may therefore harbour a mosaic of genetically different populations of *V. inaequalis*.

Fungicide resistances due to target mutations such as those against MBCs and QoIs are readily detectable phenotypic traits. It is a pertinent question for further work whether the pattern observed in the present study—a large homogeneous population in a continuous fruit production area such as the Altes Land, but very local populations outside—might be indicative of the distribution also of other genetic traits such as those relating to the adaptation of *V. inaequalis* to particular apple cultivars. An obvious example is the breakage of *Vf*-based resistance. Such changes will initially happen as the result of mutations in very few individual orchards, giving rise to a clonal population [53] which may form a reproductively isolated subpopulation within a fruit-growing area [54]. In the Altes Land, the first incidence of a breakage of *Vf* resistance was recorded in very few organic farms in 2012, and more frequent cases have been observed since 2021 (P. Heyne and R. Weber, unpublished observations). Our finding of occasional fungicide resistance in *Vf*-resistant strains within the Altes Land, but their near-absence in the periphery, indicates that a mixing of these populations may have begun in the core area. It would be very interesting to follow the fungicide resistance properties of *Vf*-breaking strains within and outside the Altes Land over time, especially in IPM orchards where cultivars with *Vf*-based resistance are also now being introduced. If a genetic exchange enables *V. inaequalis* to adapt faster to new cultivars, including those with a partial (field) resistance rather than a complete (monogenetic) resistance, large coherent apple producing regions might be hotspots for the development of more aggressive strains of *V. inaequalis*.

## 5. Conclusions

Fungicide resistance surveys such as the present one provide opportunities to examine the short-term evolution of *V. inaequalis* populations in response to conditions imposed by commercial fruit production. Selective pressures might be imposed by the types of fungicide used and their frequencies of application, but also by apple cultivars and their share of acreage in commercial orchards and by the density of apple orchards in a given area. The Altes Land core area is subject to the most intensive form of apple production conceivable. In comparison with a survey of its periphery, it permits conclusions concerning the elevated risk of fungicide resistance development in intensively managed orchards, the persistence of MBC and QoI resistance even without continued selection pressure, and the lack of spread of such strains over large distances exceeding 1 km. Further, the close proximity of organic orchards to IPM orchards and the increasing use of *Vf*-resistant apple cultivars provide opportunities for the study of *V. inaequalis* population dynamics which we aim to explore further by repeating our resistance surveys at regular intervals in future.

## Figures and Tables

**Figure 1 biotech-14-00044-f001:**
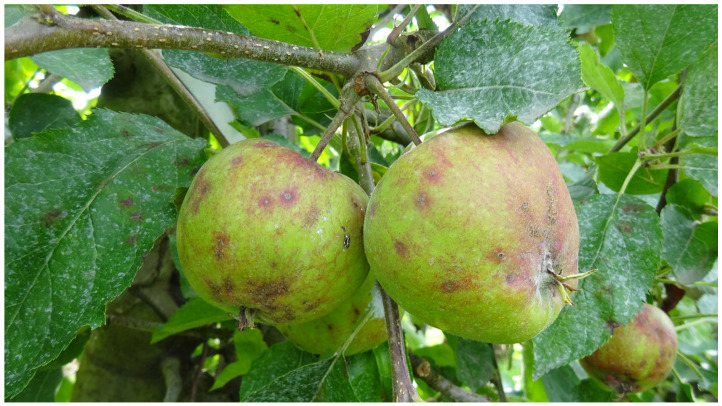
Fruit and leaf scab caused by *Venturia inaequalis* on the highly scab-susceptible cultivar ‘Jonagold’ in a commercial organic orchard from the Lower Elbe region. The photograph was taken in July 2024 following unusually wet conditions in spring and early summer. The intensity of the fungicide spray regime is indicated by white lime-sulphur deposits on the leaves and the russeting on the fruit surface as a result of repeated applications of copper-based fungicides.

**Figure 2 biotech-14-00044-f002:**
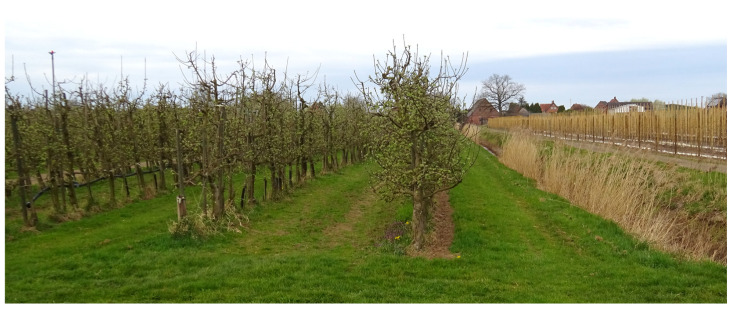
Organically managed commercial orchard in the Lower Elbe region, photographed in early spring before flowering.

**Figure 3 biotech-14-00044-f003:**
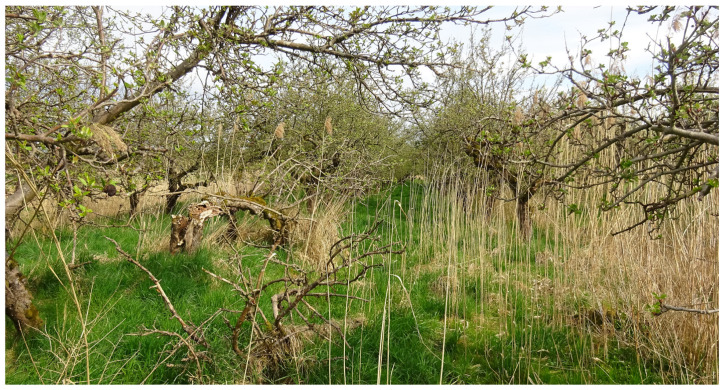
Abandoned old-style orchard with large trees on seedling rootstocks, photographed on the same day as Figure 2.

**Figure 4 biotech-14-00044-f004:**
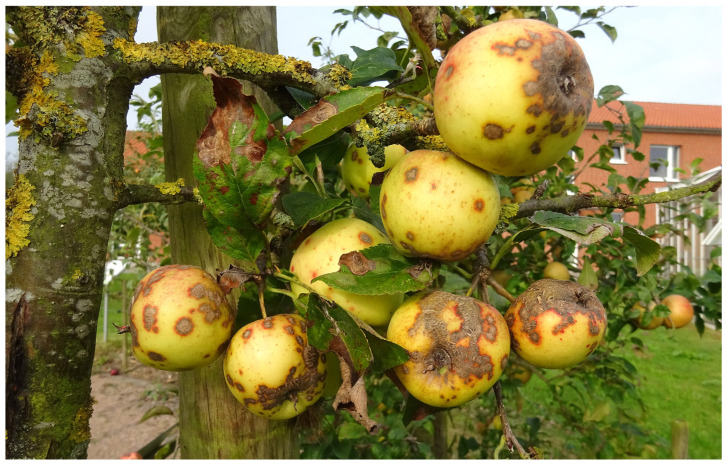
Apple tree left untreated for 10 years since its planting, showing colonisation of the bark by lichens as well as symptoms of apple scab and leaf mines of the blister moth *Leucoptera malifoliella* (Costa).

**Figure 5 biotech-14-00044-f005:**
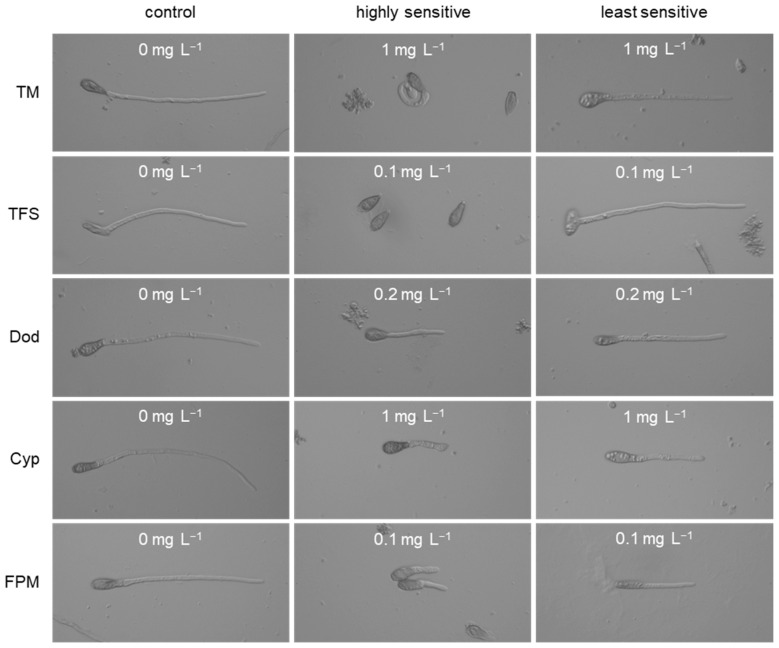
The range of growth responses of *Venturia inaequalis* to discriminatory concentrations of five fungicides, illustrated with isolate 42-14 (highly sensitive) and isolate 30-25 (least sensitive or resistant).

**Figure 6 biotech-14-00044-f006:**
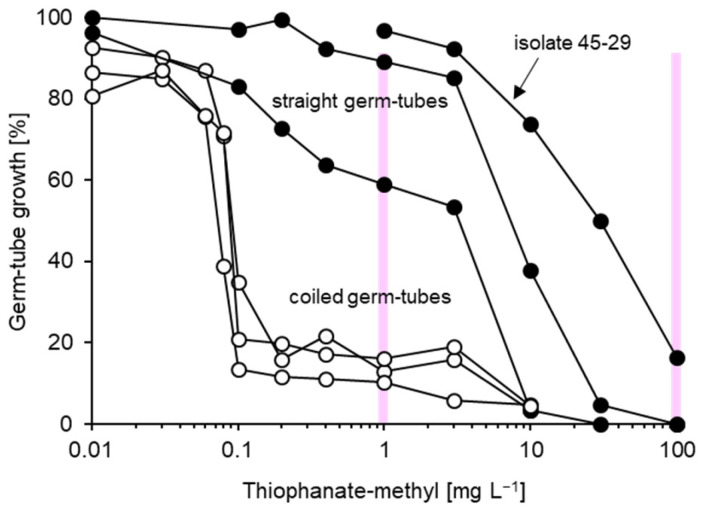
Representative thiophanate-methyl inhibition curves for three sensitive (white circles) and three resistant (black circles) isolates of *Venturia inaequalis*. At sublethal concentrations, the germ-tubes of sensitive isolates were much reduced due to hyphal coiling typical of MBC inhibition. Two discriminatory concentrations chosen for the resistance survey are indicated as vertical lines.

**Figure 7 biotech-14-00044-f007:**
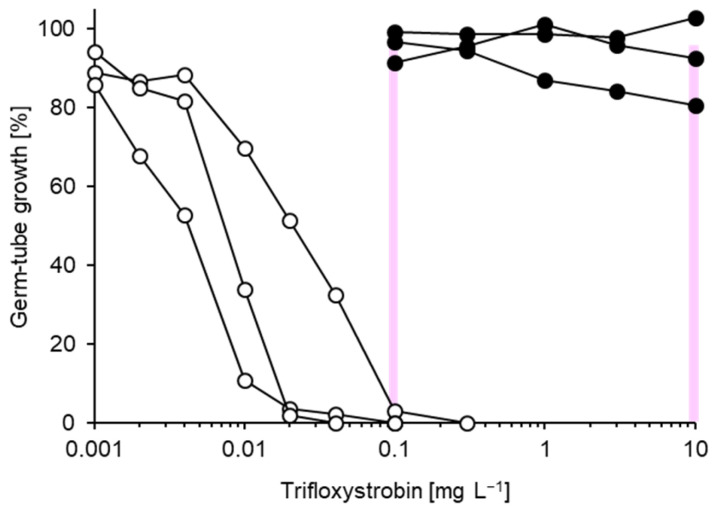
Representative trifloxystrobin inhibition curves for three sensitive (white circles) and three resistant (black circles) isolates of *Venturia inaequalis*. At sublethal concentrations, germ-tube growth of sensitive isolates stalled after germination. Two discriminatory concentrations chosen for the resistance survey are indicated as vertical lines.

**Figure 8 biotech-14-00044-f008:**
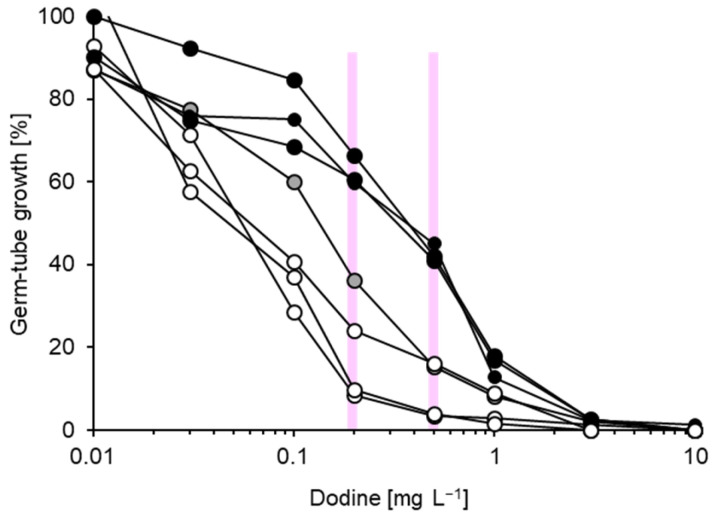
Representative dodine inhibition curves for three sensitive (white circles), one intermediate (grey circles) and three partially resistant (black circles) isolates of *Venturia inaequalis*. No qualitative distinctions between these categories were possible, necessitating measurements of germ-tube length at 0.2 and 0.5 mg dodine L^−1^ in routine assays (vertical lines). Germ-tubes were also measured at 0.1 mg dodine L^−1^, but these data were not used for analysis.

**Figure 9 biotech-14-00044-f009:**
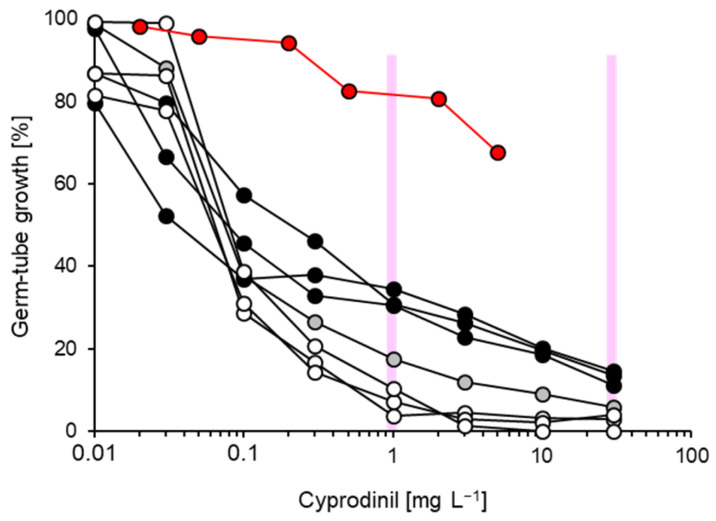
Representative inhibition curves for isolates of *Venturia inaequalis* with high (white circles), lesser (grey circles) and least (black circles) sensitivity to cyprodinil. For reference, the inhibition curve of a resistant isolate from a previous survey [21] is shown in red. Two discriminatory concentrations chosen for the resistance survey are indicated as vertical lines.

**Figure 10 biotech-14-00044-f010:**
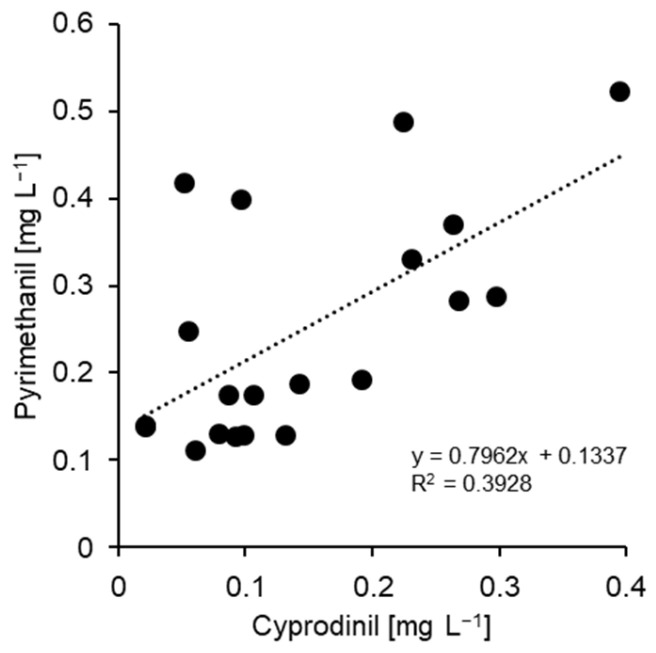
Correlation of EC_50_ values for the two AP compounds cyprodinil and pyrimthanil, obtained from detailed analysis of 20 isolates (see Table 1).

**Figure 11 biotech-14-00044-f011:**
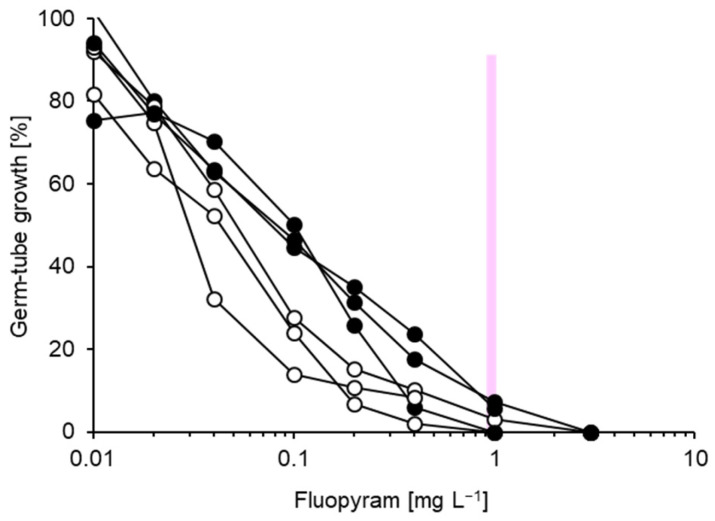
Representative inhibition curves for isolates of *Venturia inaequalis* with high (white circles) and least (black circles) sensitivity to fluopyram as observed in the present study. No evidence of resistance was obtained in this study. The lower of the two discriminatory concentrations chosen for the resistance survey is indicated as a vertical line.

**Figure 12 biotech-14-00044-f012:**
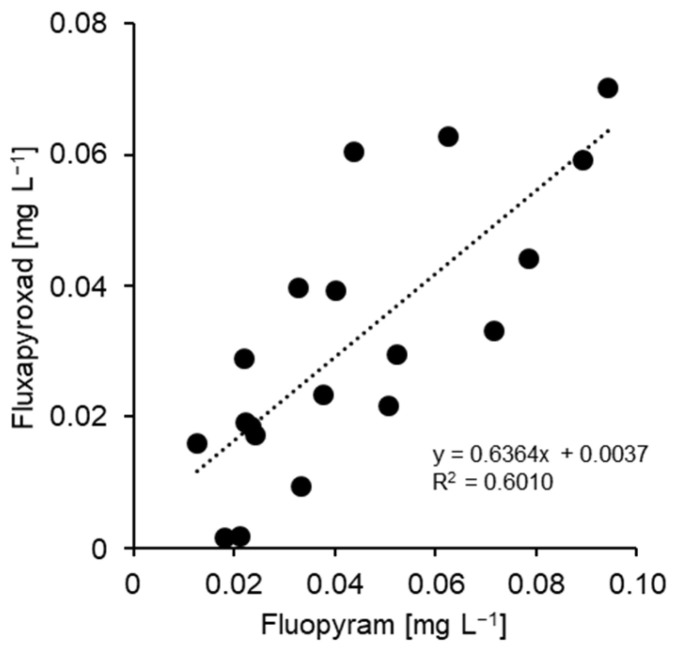
Correlation of EC_50_ values for the two SDHI compounds fluopyram and fluxapyroxad, obtained from detailed analysis of 20 isolates (see Table 1).

**Figure 13 biotech-14-00044-f013:**
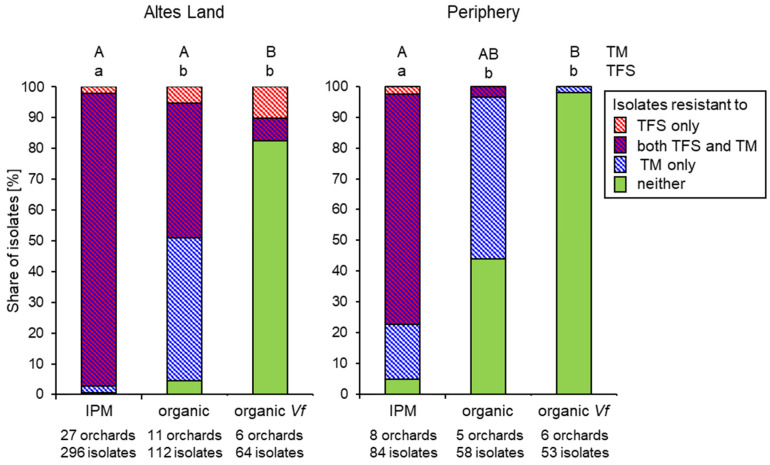
Shares of *Venturia inaequalis* isolates with resistance or sensitivity to trifloxystrobin (TFS) and thiophanate-methyl (TM) from susceptible cultivars in IPM farms, from susceptible cultivars in organic farms or from *Vf*-resistant cultivars in organic farms. Data are combined for 2021 and 2022, but separated according to the sampling areas of Altes Land and periphery. Different letters (capital for TM, lower-case for TFS) indicate significant differences between production systems within the Altes Land region and the periphery (Bonferroni, *p* < 0.05).

**Figure 14 biotech-14-00044-f014:**
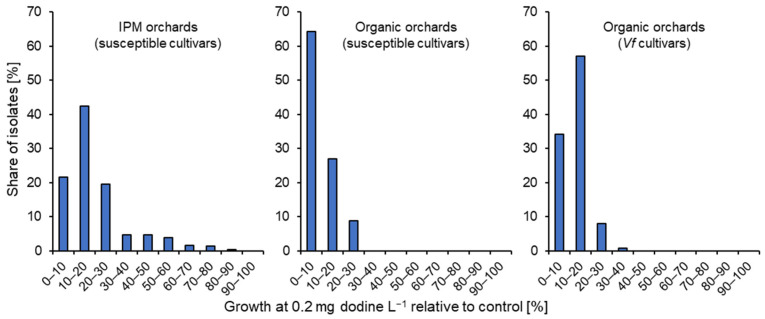
Shares of *Venturia inaequalis* isolates with different germ-tube lengths at 0.2 mg dodine L^−1^ relative to the untreated control. Pooled data from the Altes Land and periphery are presented for isolates from standard cultivars under IPM or organic management, and from *Vf* cultivars under organic management.

**Figure 15 biotech-14-00044-f015:**
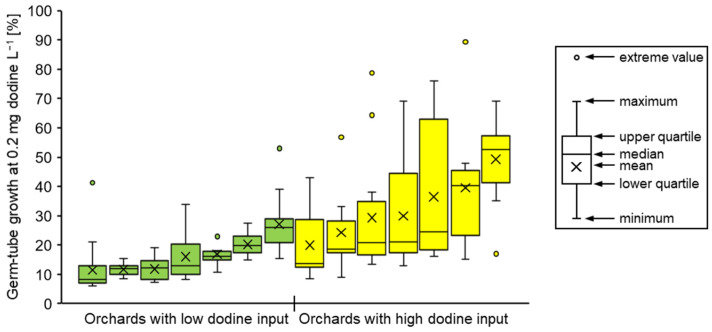
Box-plot diagrams showing germ-tube growth at 0.2 mg dodine L^−1^ for isolates from 7 IPM orchards with a known record of limited dodine use (1–2 times per season) and a further 7 IPM orchards with a known or strongly suspected use of dodine at higher annual frequencies (3–5 times per season).

**Figure 16 biotech-14-00044-f016:**
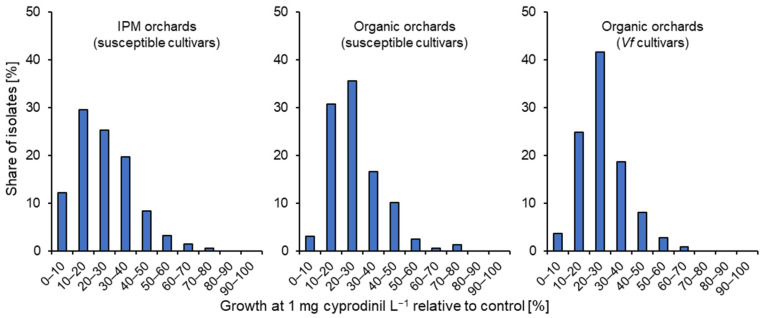
Shares of *Venturia inaequalis* isolates with different germ-tube lengths at 1 mg cyprodinil L^−1^ relative to the untreated control. Pooled data from the Altes Land and periphery are presented for isolates from standard cultivars under IPM or organic management, and from *Vf* cultivars under organic management.

**Figure 17 biotech-14-00044-f017:**
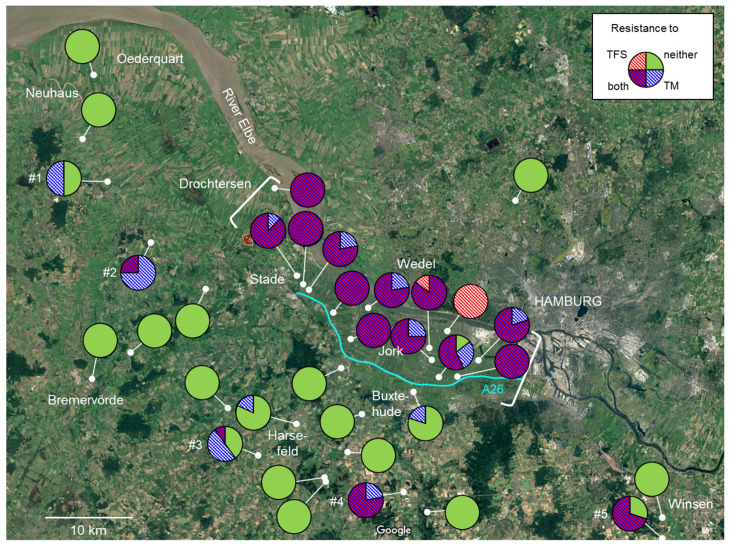
Map of the Lower Elbe region, showing the locations of abandoned orchards sampled in the periphery of and within the Altes Land core fruit production area as delimited by the river Elbe in the north, the dam of the A26 motorway (blue line) in the south and the brackets drawn to indicate the eastern and western boundaries. Locations of the periphery referred to in the text are labelled #1 to #5.

**Figure 18 biotech-14-00044-f018:**
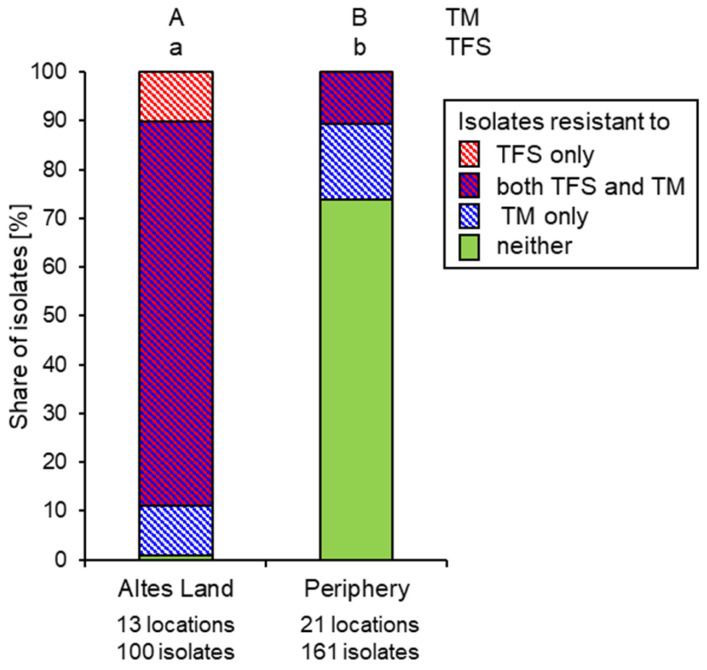
Shares of *Venturia inaequalis* strains with resistance or sensitivity to trifloxystrobin (TFS) and thiophanate-methyl (TM) in abandoned orchards and solitary trees from within the Altes Land core fruit production area, and from its periphery. Combined data for 2021 and 2022. Different letters (capital for TM, lower-case for TFS) indicate statistically significant differences between these areas (Kruskal–Wallis, *p* < 0.05).

**Table 1 biotech-14-00044-t001:** Details of *Venturia inaequalis* isolates from the Lower Elbe region and their EC_50_ values (mg active ingredient L^−1^ agar medium) towards seven fungicides tested, viz. thiophanate-methyl (TM), trifloxystrobin (TFS), dodine (Dod), cyprodinil (Cyp), pyrimethanil (Pyr), fluopyram (FPM) and fluxapyroxad (FPX).

Isolate	Cultivar	Production (Location)	TM	TFS	Dod	Cyp	Pyr	FPM	FPX
22-006	Red Jonaprince	abandoned (Borstel)	1.6846	>10	0.1133	0.0986	0.1298	0.0400	0.0394
28-26	ZIN17 (*Vf* res.)	organic (Moorende)	0.0777	>10	0.2651	0.0967	0.3997	0.0241	0.0173
28-27	ZIN17 (*Vf* res.)	organic (Moorende)	0.1126	0.0025	0.0905	0.2309	0.3318	0.0222	0.0192
28-28	ZIN17 (*Vf* res.)	organic (Moorende)	4.3362	>10	0.0732	0.0790	0.1303	0.0506	0.0218
28-29	ZIN17 (*Vf* res.)	organic (Moorende)	0.0936	0.0141	0.0778	0.0548	0.2487	0.0210	0.0019
29-25	ZIN17 (*Vf* res.)	organic (Königreich)	0.0727	0.0050	0.0451	0.0517	0.4185	0.0181	0.0017
30-25	Red Jonaprince	IPM (Rübke)	3.0845	>10	0.3521	0.2637	0.3711	0.0715	0.0333
30-27	Red Jonaprince	IPM (Rübke)	2.4043	>10	0.1644	0.3948	0.5229	0.0326	0.0399
35-18	Braeburn	IPM (Neuenkirchen)	0.0940	0.0039	0.0699	0.0608	0.1123	0.0376	0.0234
35-21	Braeburn	IPM (Neuenkirchen)	13.9703	>10	0.1376	0.1313	0.1292	0.0623	0.0628
37-03	Braeburn	IPM (Mittelnkirchen)	4.0155	>10	0.2298	0.0216	0.1411	0.0891	0.0592
37-19	Braeburn	IPM (Neuenfelde)	16.6829	>10	0.1617	0.0869	0.1749	0.0522	0.0296
38-15	Red Jonaprince	IPM (Hedendorf)	5.8886	>10	0.2584	0.1912	0.1921	0.0942	0.0702
39-21	Braeburn	IPM (Winsen)	5.3119	>10	0.0874	0.1067	0.1759	0.0490	0.0412
40-20	Jonagold	IPM (Sommerland)	0.0929	0.0065	0.0676	0.2239	0.4880	0.0331	0.0095
42-14	Topaz	organic (Osten)	0.0880	0.0055	0.0528	0.0211	0.1390	0.0234	0.0187
42-23	Topaz	organic (Osten)	0.0672	0.0064	0.0942	0.2971	0.2884	0.0126	0.0161
45-11	Red Jonaprince	IPM (Rübke)	0.1015	0.0024	0.0562	0.2681	0.2835	0.0218	0.0289
45-29	Elstar	IPM (Neuenfelde)	25.0823	>10	0.1331	0.0917	0.1278	0.0785	0.0443
45-30	Elstar	IPM (Neuenfelde)	6.9531	>10	0.0717	0.1421	0.1887	0.0438	0.0606

## Data Availability

The original contributions presented in this study are included in the article. Further inquiries can be directed to the corresponding authors.

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
