# Peer review of "Spatial and Temporal Aspects of Fungicide Resistance in Venturia inaequalis (Apple Scab) Populations in Northern Germany"

_biotech, 2025, doi:10.3390/biotech14020044_

Round 1

Reviewer 1 Report

Comments and Suggestions for Authors

Comments:

The submitted paper Spatial and temporal aspects of fungicide resistance in Venturia inaequalis (apple scab) populations in Northern Germany" by Weber, Busch and Wesche is very interesting and it is very well constructed. It is a fine contribution to the plant pathology area, in particular for the management of the apple scab fungal disease and to know more about fungicide resistance, in this particular disease, and I think it brings important results for the management of the disease.

The paper is clear, well written and well organised.

The tittle is fine to me.

The abstract is clear and embodies well the article, pointing out the main outcomes of the paper.

Also, keywords were generally well chosen, but I wouldn’t repeat those that are already present in the tittle, as for example “Venturia inaequalis”” or “apple scab"...

Moreover, I liked very much the way “Key contributions” were written.

The Introduction of the paper is very well written and well organized, and it is very complete.

The objectives of the paper are clearly indicated at the end of introduction.

The Materials and Methods section is well organized (the subsections created are useful for the article); the methods are adequately described and with detail. Also, sampling was planned in a proper way. This paper represents a huge amount of work and good science was done!

Moreover, the authors employed adequate statistics.

Results are also very well organised and properly described, with proper figures and tables. Figures and images are of good quality. I emphasize again that the paper represents a huge bulk of data and of massive work done.

Just a remark: if possible, I would add some more detailed information to the general legends of figures 6-10 (the ones of the representative inhibition curves of the pathogen sensitivity to several fungicides).

I emphasize also the importance of the results concerning subsection 3.3 (resistance in abandoned orchards).

Discussion is well constructed and articulated, covering all points of the Results section, and it is well supported by the literature. It is really a good piece of work.

The thing that I did not like: a paper like this one deserves a “Conclusions” section. I saw some recent papers of the journal and some of them have a “Conclusions” section. I think that your paper must have one too!

The list of References is very fair and enough.

In summary, this is a very good work in my opinion that deserves publication in “BioTech” journal with no doubt! Congratulations for your work and for this nice paper!

Reviewer 2 Report

Comments and Suggestions for Authors

The manuscript entitled “Spatial and temporal aspects of fungicide resistance in Venturia 2 inaequalis (apple scab) populations in Northern Germany” is described a survey of Venturia inaequalis populations from 97 orchards in the Lower Elbe region of northern Germany. The manuscript is well organized. However, statistical problems remain to be answered.

  1. Because the MIC data are highly skewed (many 0 % or 100 % growth values), applying parametric statistics (mean ± SD, one-way ANOVA/Tukey) is inappropriate. Prematurely classifying isolates as “resistant” or “sensitive” based on these tests can distort control recommendations and orchard-management guidelines. The data should be re-analyzed with non-parametric procedures—Kruskal–Wallis followed by Dunn’s multiple-comparison test—so that confidence intervals and P values are computed correctly.
  2. Several key figures (dose-response curves, discriminatory-dose histograms, box plots) are referenced but not present in the PDF. Major journals will reject a paper simply because the underlying data are not visible. The authors must embed the original graphs or at least supply box plots of the raw distributions so reviewers and readers can verify the results.
